# Information Gain Propagation: a new way to Graph Active Learning with Soft Labels

**Wentao Zhang[1], Yexin Wang[1], Zhenbang You[1], Meng Cao[2], Ping Huang[2]**
**Jiulong Shan[2], Zhi Yang[1,3], Bin Cui[1,3,4]**
[1]School of CS, Peking University [2]Apple
[3] National Engineering Laboratory for Big Data Analysis and Applications
[4]Institute of Computational Social Science, Peking University (Qingdao), China
[1]{wentao.zhang, yexinwang, zhenbangyou, liyang.cs, yangzhi, bin.cui}@pku.edu.cn
[2]{mengcao, Huang_ping, jlshan}@apple.com

## Abstract

Graph Neural Networks (GNNs) have achieved great success in various tasks, but their performance highly relies on a large number of labeled nodes, which typically requires considerable human effort. GNN-based Active Learning (AL) methods are proposed to improve the labeling efficiency by selecting the most valuable nodes to label. Existing methods assume an oracle can correctly categorize all the selected nodes and thus just focus on the node selection. However, such an exact labeling task is costly, especially when the categorization is out of the domain of individual expert (oracle). The paper goes further, presenting a soft-label approach to AL on GNNs. Our key innovations are: i) *relaxed queries* where a domain expert (oracle) only judges the correctness of the predicted labels (a binary question) rather than identifying the exact class (a multi-class question), and ii) *new criteria* of maximizing information gain propagation for active learner with relaxed queries and soft labels. Empirical studies on public datasets demonstrate that our method significantly outperforms the state-of-the-art GNN-based AL methods in terms of both accuracy and labeling cost.

## 1 Introduction

Graph Neural Networks (GNNs) have recently achieved remarkable success in various graph-based tasks, ranging from traffic networks, biology, to social networks (Zhang et al., 2020b; Wang et al., 2019; Li et al., 2019; Do et al., 2019; Wu et al., 2020b). Despite their effectiveness and popularity, GNNs typically require a large amount of labeled data to achieve satisfactory performance. However, obtaining these labels is a time-consuming, laborious, and costly process. Therefore, how to do this economically and efficiently has attracted great attention both in academia and industry. One of the most popular strategies to tackle this challenge is Active Learning (AL) (Aggarwal et al., 2014). By combining model training and node selection for labeling, AL significantly reduces labeling cost by selecting the most valuable nodes to label.

However, previous AL methods assume the hard label (namely the exact label, that is, the label specifying the exact class of the node) can always be provided by the oracle. As illustrated in Fig. 1, for any selected node, the active learners in the aforementioned methods ask questions like "which category does it *exactly* belong to?". Such queries and assumptions exceed the capability of oracle in many labeling tasks requiring domain knowledge. For example, the task in ogbn-papers100M is to leverage the citation graph to infer the labels of the arXiv papers into 172 arXiv subject areas (a single-label, 172-class classification problem). In this example, a specialist/expert in the subject areas of machine learning is incapable of labeling query instances with subject areas of finance (such as mathematical finance or computational finance), which is out of his domain knowledge.

In this paper, we propose a novel active learning paradigm for GNNs in which only soft labels are required. Two salient features of our paradigm are as follows:

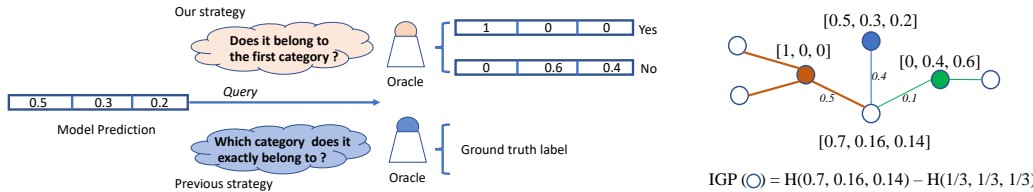

Figure 1: An example of our new node labeling strategy.   Figure 2: An example of IGP.

First, we propose relaxed queries where domain expert (oracle) only judges the correctness of the predicted labels (by GNN models) rather than the exact categorization. Specifically, we propose to select the class label with the highest predicted probability and then ask the oracle to judge whether the class label is correct rather than directly asking for the exact class label. In this way, the multi-class classification task is relaxed into a binary-class classification task for the oracle. Under such queries, if the model prediction is incorrect (we assume the oracle does not make mistakes), we annotate the node with the soft label by re-normalizing the model predicted softmax outputs over the remaining classes. As illustrated in Fig.1, we allow relaxed queries such as "Does the sample belong to this class?" (e.g., the first class in Fig. 1), since this query is specific to the domain of experts, hence more ready to be answered. As far as we are concerned, no previous AL methods can deal with such relaxed queries.

Second, the node selection criteria in previous AL methods are specially designed for hard-labeling oracles. As one of the most commonly used AL query strategies, uncertainty sampling queries the labels of the nodes which current model is least certain with, ensuring the largest entropy reduction on a single node when given the hard label from oracle. However, under our relaxed queries, we prove that the uncertainty sampling cannot guarantee the largest information gain (i.e., entropy reduction) on a single node. In this paper, we propose a new criterion to explicitly *maximize information gain propagation (IGP)* for active learners with relaxed queries and soft labels on graph. As shown in Fig. 2, the labeled nodes (with color) can propagate their label information with different influence magnitude and then reduce the uncertainty of adjacent unlabeled nodes (without color). Considering the influence propagation in GNNs, we select the nodes that can maximize expected information gain in total under the relaxed queries, i.e., reducing the aggregated uncertainty of a neighborhood region beyond a single node. To the best of our knowledge, this is the first work to propose the information gain propagation with relaxed queries, and showing that it is highly effective.

In summary, the core contributions of this work are the following: First, we provide a new AL paradigm for GNNs methodology with relaxed queries and soft labels. Second, we propose a novel node selection criterion that explicitly maximizes propagation of information gain under the new AL paradigm on GNNs. Third, experimental results show our paradigm significantly outperforms the compared baselines by a large margin in five open graph datasets.

## 2 PRELIMINARY

### 2.1 PROBLEM FORMULATION

Let $\mathcal{G} = (\mathcal{V}, \mathcal{E})$ with $|\mathcal{V}| = N$ nodes and $|\mathcal{E}| = M$ edges be the graph, and $c$ is the number of classes. $\mathbf{X} = \{\boldsymbol{x_1}, \boldsymbol{x_2}..., \boldsymbol{x_N}\}$ is the node feature matrix, the one-hot vector $\boldsymbol{y}_i \in \mathbb{R}^c$ and and $\boldsymbol{x_i} \in \mathbb{R}^d$ are the ground-truth label and node feature for node $v_i \in \mathcal{V}$, respectively. We consider a general AL setting in this work. Suppose the full node set $\mathcal{V}$ is partitioned into training set $\mathcal{V}_{train}$, validation set $\mathcal{V}_{val}$, and test set $\mathcal{V}_{test}$, and $\mathcal{V}_{train}$ can further be partitioned into the labeled set $\mathcal{V}_l$ and unlabeled set $\mathcal{V}_u$, respectively. Given the labeling size $\mathcal{B}$, the loss function $\ell$, the goal of graph-based AL is to select a subset $\mathcal{V}_l \subset \mathcal{V}_{train}$ to label, so that the model $f$ trained with the supervision of $\mathcal{V}_l$ can get the lowest loss on $\mathcal{V}_{test}$:

$$\underset{\mathcal{V}_l : |\mathcal{V}_l| = \mathcal{B}}{\arg\min} \mathbb{E}_{v_i \in \mathcal{V}_{test}} \left[ \ell \left( \boldsymbol{y}_i, \hat{\boldsymbol{y}}_i \right) \right], \tag{1}$$

where $\hat{\boldsymbol{y}}_i$ is the softmax outputs of node $v_i$ given by the model $f$.

Different from the previous setting, $\mathcal{B}$ is the labeling cost (i.e., money) rather than the size in our new scenario. As the experts are hard to annotate the node out of their domain, we assume the conventional

query cost $c - 1$ times as much as our relaxed query for each selected node. Correspondingly, the objective in our new scenario is

$$\underset{\mathcal{V}_l : M(\mathcal{V}_l) = \mathcal{B}}{\arg \min} \; \mathbb{E}_{v_i \in \mathcal{V}_{test}} \left[ \ell \left( \boldsymbol{y}_i, \hat{\boldsymbol{y}}_i \right) \right], \tag{2}$$

where $M(\mathcal{V}_l)$ is the labeling cost for annotating $\mathcal{V}_l$, and we focus on $f$ being GNNs due to their state-of-the-art performance in many semi-supervised node classification tasks.

## 2.2 GRAPH NEURAL NETWORKS

Unlike the images, text or tabular data, where training data are independently and identically distributed, samples are connected by edges in the graph, and each node in a GNN can aggregate the node embedding or feature from its neighbors to enhance their embedding along edges. Let $f\left(\mathbf{A}, \mathbf{X}^{(k)}; \mathbf{W}^{(k)}\right)$ be any GNN, where $\mathbf{A}$ is an adjacency matrix, $\mathbf{X}^{(k)}$ is the feature matrix of $k$-th layer, and $\mathbf{W}^{(k)}$ are the learned weights of $k$-th layer.

Taking the widely used Graph Convolution Network (GCN) (Kipf & Welling, 2017a) as an example, each GCN layer can be formulated as:

$$\mathbf{X}^{(k+1)} = f\left(\mathbf{A}, \mathbf{X}^{(k)}, \mathbf{W}^{(k)}\right) = \delta\left(\widetilde{\mathbf{D}}^{-1}\widetilde{\mathbf{A}}\mathbf{X}^{(k)}\mathbf{W}^{(k)}\right), \tag{3}$$

where $\mathbf{X}$ (and $\mathbf{X}^{(0)}$) is the original node feature, $\mathbf{X}^{(k)}$ and $\mathbf{X}^{(k+1)}$ are the embeddings of layer $k$ and $k+1$ respectively. Besides, $\widetilde{\mathbf{D}}$ is the diagonal node degree matrix for normalization and $\widetilde{\mathbf{A}} = \mathbf{A} + \mathbf{I}_N$ is the adjacent matrix with self connection, where $\mathbf{I}_N$ is the identity matrix. Compared with the DNN layer which formulated as $\mathbf{X}^{(k+1)} = \delta\left(\mathbf{X}^{(k)}\mathbf{W}^{(k)}\right)$, the feature matrix $\mathbf{X}^{(k)}$ will be firstly enhanced with the feature propagation operation $\widetilde{\mathbf{D}}^{-1}\widetilde{\mathbf{A}}\mathbf{X}^{(k)}$, and then boosts the semi-supervised node classification performance by getting more nodes involved in the model training.

## 2.3 ACTIVE LEARNING

**Common AL.** AL can improve labeling efficiency by selecting the most valuable samples to label. Considering the informativeness, Uncertainty Sampling (Yang et al., 2015; Zhu et al., 2008) selects the nodes which has the most uncertain model prediction. Based on ensembles (Zhang et al., 2020a), Query-by-Committee (Burbidge et al., 2007; Melville & Mooney, 2004) trains a committee of models, and selects samples according to the extent to which the models disagree. Furthermore, Density-based (Zhu et al., 2008; Tang et al., 2002) and Clustering-based (Du et al., 2015; Nguyen & Smeulders, 2004) and Diversity-based (Wu et al., 2020a; Jiang & Gupta, 2021) methods can effectively select the most representative samples. All the methods above are proposed for binary or multi-class classification, and some methods (Qi et al., 2008; Yan & Huang, 2018) are proposed for the multi-label classification problem. Besides, considering the class imbalance, some methods (Aggarwal et al., 2020; Attenberg & Provost, 2010) are also proposed to tackle this issue with AL.

**GNN-based AL.** Despite the effectiveness of common AL methods, it is unsuitable to directly apply them to GNNs since the characteristic of influence propagation has not been considered. To tackle this issue, both AGE (Cai et al., 2017) and ANRMAB (Gao et al., 2018) introduce the density of node embedding and PageRank centrality into the node selection criterion. SEAL (Li et al., 2020) devises a novel AL query strategy in an adversarial way, and RIM (Zhang et al., 2021b) considers the noisy oracle in the node labeling process of graph data. Besides, ALG (Zhang et al., 2021a) proposes to maximize the effectiveness of all influenced nodes of GNNs. Recently, Grain (Zhang et al., 2021c) introduces a new diversity influence maximization objective, which takes the diversity of influenced nodes into consideration.

Both AGE (Cai et al., 2017) and ANRMAB (Gao et al., 2018) are designed for AL on GNNs, which adopt the uncertainty, density, and node degree to select nodes. ANRMAB improves AGE by introducing a multi-armed bandit mechanism for adaptive decision-making. Considering the high training time of GNN, ALG (Zhang et al., 2021a) decouples the GNN model and proposes a new node selection metric that maximizes the effective reception field. Grain (Zhang et al., 2021c) further generalizes the reception field to the number of activated nodes in social influence maximization and introduces the diversity influence maximization for node selection. However, all these AL methods

assume the oracle knows the information of all categories, thereby being able to correctly annotate all the selected nodes. This is impractical if the oracle encounters some areas that they are not familiar with, especially when the number of categories is large. IGP is the first work to actively labeling graph data under domain-specific experts.

## 3 PROPOSED METHOD

This section presents IGP, the first graph-based AL framework that considers the relaxed queries and soft labels. IGP can operate on any GNN model $f$. We measure the information gain of selecting a single node in Sec. 3.1. To extend the information gain of a single node to the full graph, we firstly estimate the influence magnitude of graph propagation in Sec. 3.2, and then combine it with the information gain and introduce the new criterion: information gain propagation in Sec. 3.3. Last, we propose to select nodes which can maximize the information gain propagation in Sec. 3.4.

### 3.1 INFORMATION GAIN

Different from the previous annotation method, which asks the oracle to directly provide annotated class $l$, the oracle in IGP only needs to judge whether the model prediction $\hat{y}$ is correct. Rather than discard the incorrect prediction, we propose that both the correct and incorrect predictions will provide the information gain and thus enhance the label supervision.

An oracle is required to answer whether the model prediction is correct, i.e., $\hat{y}$ belongs to class $l$. After the judgement of the oracle, the uncertainty of the node labels will be decreased. Therefore, we define the normalized soft label by combining the model prediction and oracle judgement as follows.

**Definition 3.1 (Normalized Label).** *For node $v_i$, the soft label (model predicted softmax output) is denoted as $\hat{y}_i = [a_1, a_2, ..., a_c]$. The pseudo label (denoted as $l$) is set as the class with the highest value in $\hat{y}_i$. After the oracle judges whether $v_i$ belongs to class $l$, each element $a'_t$ in the normalized soft label $\hat{y}'_i = [a'_1, a'_2, ..., a'_c]$ is defined as*

$$a'_t = \begin{cases} \mathbb{1}(t=l) & v_i \text{ belongs to class } l \\ \frac{a_t \cdot \mathbb{1}(t=l)}{\sum_{j \neq l} a_j} & v_i \text{ does not belong to class } l \end{cases} \tag{4}$$

After the query, $\hat{y}_i$ is updated as the normalized soft label $\hat{y}'_i$ by Eq. 4 (See more details in Appendix B).

The oracle may agree with the pseudo label (denoted as $v_i+$), or disagree (denoted as $v_i-$). In either cases, the entropy drops.

The expected information gain (IG) is calculated based on these two cases.

**Definition 3.2 (Information Gain (IG)).** *For node $v_i$, the IG of annotating it is defined as*
$$IG(v_i) = H(\hat{y}_i) - P(v_i-)H(\hat{y}'_i, v_i-), \tag{5}$$
*where $H$ is the entropy (well-known in the information theory), e.g., $H(\hat{y}_i) = -\sum_{p=1}^{k} a_p \log a_p$ ($\log$ always stands for $\log_2$ in this paper). Therefore, $H(\hat{y}_i)$ and $H(\hat{y}'_i)$ stand for the entropy of $v_i$ before and after annotation respectively. Also, $H(\hat{y}'_i, v_i+)$ and $H(\hat{y}'_i, v_i-)$ are the two cases of $H(\hat{y}'_i)$ where $v_i+$ and $v_i-$ occurs respectively.*

IG is exactly **the expectation of entropy reduction**, as proved in Appendix C.

The confidence of the model in the pseudo label is shown in the soft label. Hence, before the annotation, this is a good measurement for the probability that the oracle will agree with the pseudo label, namely $P(v_i+)$.

Here following theorem shows the difference between IG and entropy.

**Theorem 3.1.** *The node with the highest entropy is not necessarily the node that brings the highest IG.*

The proof of this theorem can be found in Appendix A.

More examples about IG in differing cases, that is, the oracle may agree or disagree with the pseudo label, can be found in Appendix D.

## 3.2 Influence Magnitude Estimation

Each newly annotated node will propagate its label information to its $k$-hop neighbors in a $k$-layer GNN and correspondingly influences the label distribution of these nodes. Since the influence of a node on its different neighbors can be various, we measure it by calculating how much change in the input label of node $v_i$ affects the *propagated* label of node $v_j$ after $k$-steps propagation (Wang & Leskovec, 2020; Xu et al., 2018).

**Definition 3.3** (**Influence Magnitude**). *The influence magnitude score of node $v_i$ on node $v_j$ after $k$-step propagation is the L1-norm of the expected Jacobian matrix $\hat{I}_f(v_j, v_i, k) = \left\| \mathbb{E}[\partial \mathbf{X}_j^{(k)} / \partial \mathbf{X}_i^{(0)}] \right\|_1$. Formally, we normalize the influence magnitude score as:*

$$I_f(v_j, v_i, k) = \frac{\hat{I}_f(v_j, v_i, k)}{\sum_{v_w \in \mathcal{V}} \hat{I}_f(v_j, v_w, k)}. \tag{6}$$

For a $k$-layer GNN, the influence magnitude score $I_f(v_j, v_i, k)$ captures the sum over probabilities of all possible influential paths with length of $k$ from node $v_i$ to $v_j$. Intuitively, node $v_i$ will influence $v_j$ by a larger extent and thus $I_f(v_j, v_i, k)$ is larger if $v_i$ is easier to achieve at $v_j$ after the random walk. In our AL setting, larger $I_f(v_j, v_i, k)$ means the label distribution of node $v_j$ will be influenced more by node $v_i$ if $v_i$ is labeled.

## 3.3 Information Gain Propagation

For a $k$-layer GNN, the supervision signal of one node can be propagated to its $k$-hop neighborhood nodes and then influences the label distribution of these neighbors. Therefore, it is unsuitable to just consider the information gain of the node itself in the node selection process. So, we extend the information gain of a single node defined in Def. 3.2 to its $k$-hop neighbors in the graph. We firstly combine the influence magnitude with the normalized soft label and then measure the information gain propagation of all the influenced nodes. Concretely, the information gain propagation of node $v_j$ from the labeled node $v_i$ is defined as follows.

**Definition 3.4** (**information gain propagation**). *Given the model predicted softmax outputs $\hat{\boldsymbol{y}}_i$ of node $v_i$, the influence magnitude $I_f(v_j, v_i, k)$ of node $v_i$ on $v_j$, the information gain propagation of node $v_j$ from labeling node $v_i$ is defined as:*

$$IGP(v_j, v_i, k) = H\left( \sum_{v_m \in \mathcal{V}_l} I_f(v_j, v_m, k) \hat{\boldsymbol{y}}_m' \right) - H\left( \sum_{v_m \in \mathcal{V}_l \cup \{v_i\}} I_f(v_j, v_m, k) \hat{\boldsymbol{y}}_m' \right). \tag{7}$$

Since the new soft label $\hat{\boldsymbol{y}}_i'$ is unavailable for node $v_i$ before the annotation, the expected information gain propagation (IGP) is calculated by

$$IGP(v_j, v_i, k) = P(v_i+) IGP(v_j, v_i, k, v_i+) + P(v_i-) IGP(v_j, v_i, k, v_i-), \tag{8}$$

where $IGP(v_j, v_i, k, v_i+)$ is the IGP when the oracle agrees with the pseudo label. In this case, $\hat{\boldsymbol{y}}_i'$ is a one-hot vector according to E.q. 4.

## 3.4 Maximizing Information Gain Propagation

To maximize the uncertainty of all the influenced nodes in the semi-supervised GNN training, we aim to select and annotate a subset $\mathcal{V}_l$ from $\mathcal{V}$ so that the maximum information gain propagation can be achieved according to Eq. 9. Specifically, we optimize this problem by defining the information gain propagation maximization objective as below.

**Objective.** Specifically, we propose to maximize the following objective:

$$\max_{\mathcal{V}_l} F(\mathcal{V}_l) = \sum_{v_i \in \mathcal{V}_l} \sum_{v_j \in RF(v_i)} IGP(v_j, v_i, k), \text{s.t. } \mathcal{V}_l \subseteq \mathcal{V}, |\mathcal{V}_l| = \mathcal{B}. \tag{9}$$

where $RF(v_i)$ is the receptive field of node $v_i$, i.e., the node itself with its $k$-hop neighbors in a $k$-layer GNN. Considering the influence propagation, the proposed objective can be used to find a subset $\mathcal{V}_l$ that can maximize the information gain of all influenced nodes as more as possible.

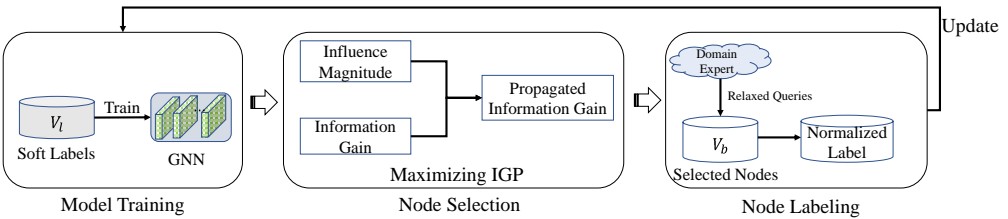

Figure 3: An overview of the proposed IGP framework.

---

**Algorithm 1:** Working pipeline of IGP

---

**Input:** Initial labeled set $\mathcal{V}_0$, query batch size $b$.
**Output:** Labeled set $\mathcal{V}_l$
1 **Stage 1: Node Selection**
2 $\mathcal{V}_l = \mathcal{V}_0, \mathcal{V}_b = \emptyset$;
3 **for** $t = 1, 2, \ldots, b$ **do**
4 $\quad$ Select the most valuable node: $v_i = \arg\max_{v \in \mathcal{V}_{train} \setminus \mathcal{V}_l} F(\mathcal{V}_l \cup \{v\})$;
5 $\quad$ $\mathcal{V}_l = \mathcal{V}_l \cup \{v_i\}, \mathcal{V}_b = \mathcal{V}_b \cup \{v_i\}, \mathcal{V}_{train} = \mathcal{V}_{train} \setminus \{v_i\}$;
6 **Stage 2: Node Labeling**
7 **for** $v_i \in \mathcal{V}_b$ **do**
8 $\quad$ Judge the correctness of the predicted pseudo label $\bar{y}_i$;
9 $\quad$ **if** $\bar{y}_i$ *is correct (agreed by the oracle)* **then**
10 $\quad\quad$ Labeling $v_i$ with $\hat{\boldsymbol{y}}_i'$ according to Eq. 4;
11 $\quad$ **else**
12 $\quad\quad$ Labeling $v_i$ with $\hat{\boldsymbol{y}}_i'$ according to Eq. 4;
13 $\quad\quad$ $\mathcal{V}_l = \mathcal{V}_l \setminus \{v_i\}, \mathcal{V}_{train} = \mathcal{V}_{train} \cup \{v_i\}$;
14 **Return** $\mathcal{V}_l$

---

## 4 IGP FRAMEWORK

### 4.1 FRAMEWORK OVERVIEW

As shown in Fig. 3, for each batch of node selection, IGP first trains the GNN model with the soft labels. With the model prediction, IGP measures the expected information gain of labeling each node. Considering both the influence magnitude and information gain, IGP further measures the information gain propagation of each unlabeled node and selects a batch of nodes that can maximize the information gain propagation on the graph. After that, with the relaxed queries, the oracle only judges the correctness of the model prediction, and the label distribution will be normalized correspondingly. At last, we re-train the GNN model with update the model prediction. The above process is repeated until the labeling cost $\mathcal{B}$ runs out.

### 4.2 WORKING PIPELINE

**Node Selection.** Without losing generality, consider a batch setting where $b$ nodes are selected in each iteration. Algorithm 1 provides a sketch of our greedy selection method for GNNs. Given the training set $\mathcal{V}_{train}$, and query batch size $b$, we first select the node $v_i$, which generates the maximum marginal gain (Line 3). We then update the labeled set $\mathcal{V}_l$, the oracle annotated set $\mathcal{V}_b$, and the training set $\mathcal{V}_{train}$ (Line 4). This process is repeated until a batch of $b$ nodes have been selected. Note that the normalized soft label $\hat{\boldsymbol{y}}_i'$ is unavailable before the node labeling process. For better efficiency, we assume $\hat{\boldsymbol{y}}_i'$ is the model predicted soft label $\hat{\boldsymbol{y}}_i$ in the node selection process. If $\hat{\boldsymbol{y}}_i$ is not confident (i.e.,[0.4,0.3,0.3] for a three-classification problem), it will be of little influence on the label distribution of its neighbors. On the contrary, if $\hat{\boldsymbol{y}}_i$ is closer to $\hat{\boldsymbol{y}}_i'$ if it is confident. Consequently, it is reasonable to make such an assumption.

**Node Labeling** After getting a batch of the selected node-set $\mathcal{V}_b$, we acquire the label from an oracle. As introduced before, different from previous works which require the oracle to annotate the hard

label directly, the oracle in our framework only needs to judge the correctness of the pseudo label $\bar{y}_i$ (Line 8). If $\bar{y}_i$ is correct, we update its soft label $\hat{y}_i'$ according to Eq. 4 (Line 10). Note that we can still get information gain even if $\bar{y}_i$ is incorrect. For any node $v_i$ with an incorrect pseudo label, we firstly update its soft label $\hat{y}_i'$ (Line 10). After that, we remove it from $\mathcal{V}_l$ and add it to $\mathcal{V}_{train}$ (Line 13). In this way, this node has the chance to be re-selected in the next batch. Note that the model will produce a different label given the fact that the oracle rejects the previously predicted label.

**Model Training** Intuitively, a node with smaller entropy contains contributes more to the training process. In other words, a one-hot label contributes most, while other normalized soft labels also provide weak supervision. For GNN model training, we use the weighted loss function as follows

$$\mathcal{L} = - \sum_{v_i \in \mathcal{V}_l, H(v_i)=0} \hat{y}_i' \log \hat{y}_i + \alpha \sum_{v_i \in \mathcal{V}_l, H(v_i) \neq 0} \hat{y}_i' log \frac{\hat{y}_i'}{\hat{y}_i}, \tag{10}$$

where $\alpha$ is a hyper-parameter which controls the importance of the incorrectly predicted nodes by the model. Note that the labeled set has two types of labels: 1) one-hot hard labels (i.e., $H(v_i) = 0$), which is correctly predicted by the model; 2) soft labels (i.e., $H(v_i) \neq 0$), which has incorrect model prediction but also provides weak supervision. We adopt the KL divergence as the loss function to learn from the weak supervised soft label in both cases (cross-entropy is just a special case of KL divergence). Note that KL divergence measures the distance between the ground truth and the current prediction, and it is especially appropriate for methods with a basis of information theory.

## 4.3 EFFICIENCY OPTIMIZATION

By leveraging some recent works on scalable and parallelizable social influence maximization (Aral & Dhillon, 2018), we could enable IGP to efficiently deal with large-scale graphs. The key idea is to identify and dismiss uninfluential nodes to dramatically reduce the amount of computation for evaluating the information gain propagation. For example, we can use the node degree or the distribution of random walkers throughout the nodes (Kim et al., 2017) to filter out a vast number of uninfluential nodes.

## 5 EXPERIMENTS

### 5.1 EXPERIMENTAL SETTINGS

**Datasets.** We evaluate IGP in both inductive and transductive settings (Hamilton et al., 2017): three citation networks (i.e., Citeseer, Cora, and PubMed) (Kipf & Welling, 2017a), one large social network (Reddit), and one OGB dataset (ogbn-arxiv).

**GNN Models.** We conduct experiments using the widely used GCN model and demonstrate the generalization of IGP on other GNNs such as SGC (Wu et al., 2019), APPNP (Klicpera et al., 2019) and MVGRL (Hassani & Khasahmadi, 2020).

**Baselines.** We compare IGP with the following baselines: Random, AGE (Cai et al., 2017), ANRMAB (Gao et al., 2018), GPA (Hu et al., 2020), SEAL (Li et al., 2020), ALG (Zhang et al., 2021a), and GRAIN (Zhang et al., 2021c)

**Settings.** For each method, we use the hyper-parameter tuning toolkit (Li et al., 2021a;b) or follow the original papers to find the optimal hyper-parameters. To eliminate randomness, we repeat each method ten times and report the mean performance. Specifically, for each method, we chooses a small set of labeled nodes as an initial pool – two nodes are randomly selected for each class.

### 5.2 PERFORMANCE COMPARISON

**End-to-end Comparison.** We choose the labeling budget as 20 labels per class to show the end-to-end accuracy. Table 1 shows that GRAIN, ALG, AGE, and ANRMAB outperform Random in all the datasets, as they are specially designed for GNNs. GRAIN and ALG perform better than other baselines except for IGP because they consider the RF. However, IGP further boosts the accuracy by a significant margin. Remarkably, IGP improves the test accuracy of the best baseline, i.e., GRAIN, by 1.6-2.2% on the three citation networks, 0.9% on Reddit, and 0.6% on ogbn-arxiv.

Table 1: The test accuracy (%) on different datasets with the same labeling budget.

| Method | Cora | Citeseer | PubMed | Reddit | ogbn-arxiv |
|--------|------|----------|--------|--------|------------|
| Random | 78.8($\pm$0.8) | 70.8($\pm$0.9) | 78.9($\pm$0.6) | 91.1($\pm$0.5) | 68.2($\pm$0.4) |
| AGE | 82.5($\pm$0.6) | 71.4($\pm$0.6) | 79.4($\pm$0.4) | 91.6($\pm$0.3) | 68.9($\pm$0.3) |
| ANRMAB | 82.4($\pm$0.5) | 70.6($\pm$0.6) | 78.2($\pm$0.3) | 91.5($\pm$0.3) | 68.7($\pm$0.2) |
| GPA | 82.8($\pm$0.4) | 71.6($\pm$0.4) | 79.9($\pm$0.5) | 91.8($\pm$0.2) | 69.2($\pm$0.3) |
| SEAL | 83.2($\pm$0.5) | 72.1($\pm$0.4) | 80.3($\pm$0.4) | 92.1($\pm$0.3) | 69.5($\pm$0.1) |
| ALG | 83.6($\pm$0.6) | 73.6($\pm$0.5) | 80.9($\pm$0.3) | 92.4($\pm$0.3) | 70.1($\pm$0.3) |
| GRAIN | 84.2($\pm$0.3) | 74.2($\pm$0.3) | 81.8($\pm$0.2) | 92.5($\pm$0.1) | 70.3($\pm$0.2) |
| IGP | **86.4($\pm$0.6)** | **75.8($\pm$0.3)** | **83.6($\pm$0.5)** | **93.4($\pm$0.2)** | **70.9($\pm$0.3)** |

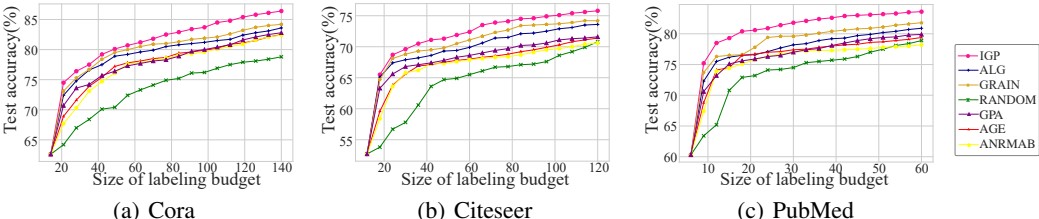

|     |     |     |
|-----|-----|-----|
| (a) Cora | (b) Citeseer | (c) PubMed |

Figure 4: The test accuracy across different labeling budgets for model training.

**Accuracy under Different Labeling Budgets.** To show the influence of the labeling budget, we test the accuracy of different AL methods under different labeling budgets on three citation datasets. Concretely, we range the budgets of oracle labels from $2C$ to $20C$ ($C$ is the number of classes) and show the test accuracy of GCN in Figure 4. The experimental results demonstrate that with the increase of labeling budget, the accuracy of IGP grows fairly fast, thereby outperforming the baselines with an even greater margin.

**Generalization.** In addition to GCN, IGP can also be applied to a large variety of GNN variants. GCN, SGC, MVGRL, and APPNP are four representative GNNs (Chen et al., 2021) which adopt different message passings. Unlike the coupled GCN, both SGC and APPNP are decoupled, while their orderings when doing feature propagation and transformation are different. Besides, MVGRL is a classic self-supervised GNN. We test the generalization ability of IGP by evaluating the aforementioned four types of GNNs on $20C$ ($C$ is the number of classes) nodes selected by IGP and other baselines in the AL scenario, and the corresponding results are shown in Table 2. The results suggest that IGP consistently outperforms the other baselines, regardless of whether the GCC is decoupled. Moreover, the result also shows our proposed method IGP can significantly outperform the compared baselines on the top of more sophisticated self-supervised GNN such as MVGRL (Hassani & Khasahmadi, 2020). As shown in the table, the test accuracy of IGP could outperform ALG and GRAIN by more than 1.8% on PubMed. Therefore, we draw a conclusion that IGP can generalize to different types of GNNs well.

**Ablation study.** IGP combines informative selecting, informative training, and information quantity in the method. To verify the necessity of each component, we evaluate IGP on GCN by disabling one component at a time. We evaluate IGP: *(i)* without the node labeled with soft label when training (called "No Informative Training (IT)"); *(ii)* without the information gain propagation when selecting nodes (called "No Informative Selection (IS)"); *(iii)* without the influence magnitude and regarding the information among all the nodes as the same (called "No Information Quantity (IM)");*(iv)* without the normalized soft label and set the soft label uniformly (called "No Normalized Label (NL)"); Table 3 displays the results of these three settings, and from the table we can learn the following information. 1) If informative training is ignored, the test accuracy will decrease in all three datasets, e.g., the accuracy gap is as large as 2.9% on PubMed. This is because informative training help utilizing the information of the nodes labeled by probability. 2) Lacking information selecting will lead to worse accuracy, since the information gain of the nodes can not only benefit itself but also its neighbors in the graph. It is more important than other components since removing the information selecting will lead to a more substantial accuracy gap. For example, the gap on Cora is 2.5%, which is higher than the other gap (2.3%). 3) Information quantity also significantly impacts model accuracy on all datasets, because it will lead to a wrong preference on the nodes with larger degree, although some of them have little influence on other nodes. 4)If we remove normalized label and set the soft

Table 2: Test accuracy of different models on PubMed.

| Method | SGC | APPNP | GCN | MVGRL |
|--------|-----|-------|-----|-------|
| Random | 77.6($\pm$0.8) | 79.2($\pm$0.6) | 78.9($\pm$0.6) | 79.3($\pm$0.5) |
| AGE | 78.8($\pm$0.5) | 79.9($\pm$0.5) | 79.4($\pm$0.4) | 79.9($\pm$0.4) |
| ANRMAB | 77.8($\pm$0.4) | 78.7($\pm$0.5) | 78.2($\pm$0.3) | 78.9($\pm$0.3) |
| GPA | 79.5($\pm$0.6) | 80.2($\pm$0.4) | 79.9($\pm$0.5) | 80.4($\pm$0.3) |
| SEAL | 79.8($\pm$0.5) | 80.5($\pm$0.5) | 80.3($\pm$0.4) | 80.7($\pm$0.3) |
| ALG | 80.5($\pm$0.4) | 81.2($\pm$0.5) | 80.9($\pm$0.3) | 81.3($\pm$0.2) |
| GRAIN | 81.1($\pm$0.3) | 82.0($\pm$0.4) | 81.8($\pm$0.2) | 82.1($\pm$0.1) |
| IGP | **83.2**($\pm$**0.6**) | **83.7**($\pm$**0.5**) | **83.6**($\pm$**0.5**) | **83.9**($\pm$**0.4**) |

Table 3: Influence of different components in test accuracy(%).

| Method | Cora | $\Delta$ | Citeseer | $\Delta$ | PubMed | $\Delta$ |
|--------|------|----------|----------|----------|--------|----------|
| No IT | 84.1($\pm$0.7) | -2.3 | 73.2($\pm$0.3) | -2.6 | 80.7($\pm$0.5) | -2.9 |
| No IS | 83.9($\pm$0.5) | -2.5 | 73.1($\pm$0.4) | -2.7 | 80.4($\pm$0.4) | -3.2 |
| No IM | 84.5($\pm$0.6) | -1.9 | 73.6($\pm$0.5) | -2.2 | 81.5($\pm$0.6) | -2.1 |
| No NL | 84.4($\pm$0.5) | -2.0 | 73.4($\pm$0.3) | -2.4 | 81.3($\pm$0.4) | -2.3 |
| **IGP** | **86.4**($\pm$**0.6**) | – | **75.8**($\pm$**0.3**) | – | **83.6**($\pm$**0.5**) | – |

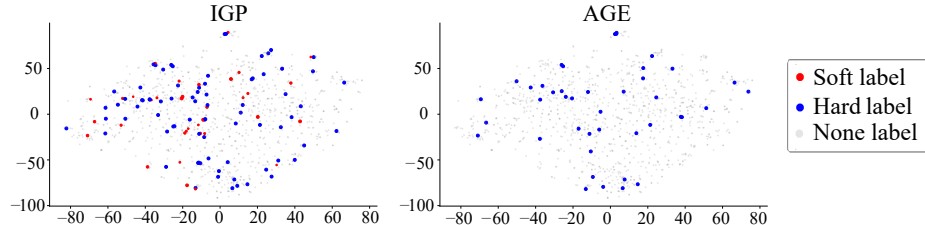

Figure 5: The node distribution with different labels of different methods on the Cora dataset.

label uniformly, the test accuracy will decrease apparently. Although the performance degradation is smaller than no IT and no IS, the results still show the necessity of normalized label.

**Interpretability** To show the insight of IGP, we evaluate and draw the distribution of the nodes with soft labels (in red color) and hard labels (in blue color), also those without labels (in gray color) for two methods: AGE and IGP when the labeling budget is $6C$ for Cora in GCN, the more information entropy the nodes have, the smaller its radium (one-hot labeled node has the largest radium). The result in Figure 5 shows that compared to AGE, IGP has enough labeled nodes with a balanced distribution, both in probability and one-hot, which could explain the better performance of NC-ALG in node classification. We also show the influence on information entropy in Appendix.

## 6 CONCLUSION

This paper presents Informatioin Gain Propagation (IGP), the first GNN-based AL method which explicitly considers AL with soft labels. Concretely, we provide a new way for active learners to exhaustively label nodes with a limited budget, using a combination of domain experts and GNN model. IGP allows us to use *relaxed queries* where domain expert (oracle) only judges the correctness of the predicted labels rather than the exact categorization, and ii) *new criteria* of maximizing information gain propagation for the active learner with relaxed queries and soft labels. Empirical studies on real-world graphs show that our approach outperforms competitive baselines by a large margin, especially when the number of classes is large.

# 7    REPRODUCIBILITY

The source code of IGP can be found in Github (`https://github.com/zwt233/IGP`). To ensure reproducibility, we have provided the overview of datasets in Section F.1 and Table 4 in Appendix F. The detailed hyperparameter settings for our IGP can be found in Appendix F.2. Our experimental environment is presented in Appendix F.2, and please refer to "README.md" in the Github repository for more details.

## ACKNOWLEDGMENTS

This work is supported by NSFC (No. 61832001, 61972004), Beijing Academy of Artificial Intelligence (BAAI), and PKU-Tencent Joint Research Lab. Bin Cui is the corresponding author.

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

## A  PROOFS

**Theorem A.1.** *The node with highest entropy (in terms of the soft label given by the model) is not necessarily the node that brings highest expected information gain.*

Let us consider a three classification problem. Assume there is a node $s$ with soft label $[1 - 2a, a, a]$, where $0 \leq a \leq \frac{1}{3}$ so that $1 - 2a \geq a$. Therefore, the entropy of $s$ is given by

$$H(s) = -(1 - 2a)\log(1 - 2a) - a\log a - a\log a = -(1 - 2a)\log(1 - 2a) - 2a\log a \quad (11)$$

When the oracle gives a negative answer, the soft label will become $[0, 0.5, 0.5]$, and thus the entropy will become $-0.5 \log 0.5 - 0.5 \log 0.5 = \log 2 = 1$. Consequently, the expected information gain of $s$ is given by

$$IG(s) = (1 - 2a)(H(s) - 0) + 2a(H(s) - 1) = H(s) - 2a \tag{12}$$

Take the derivatives of the above two variables, we have

$$
\begin{aligned}
\frac{\mathrm{d}H(s)}{\mathrm{d}a} &= [-(1 - 2a)\log(1 - 2a) - 2a \log a]' \\
&= -\{[(1 - 2a)\log(1 - 2a)]' + 2(a \log a)'\} \\
&= -\left\{[\log(1 - 2a) + \frac{1}{\ln 2}] * (-2) + 2(\log a + \frac{1}{\ln 2})\right\} \\
&= 2[\log(1 - 2a) - \log a] \\
&= 2 \log \frac{1 - 2a}{a}
\end{aligned}
\tag{13}
$$

$$
\begin{aligned}
\frac{\mathrm{d}IG(s)}{\mathrm{d}a} &= \frac{\mathrm{d}(H(s) - 2a)}{\mathrm{d}a} \\
&= \frac{\mathrm{d}H(s)}{\mathrm{d}a} - 2 \\
&= 2 \log \frac{1 - 2a}{a} - 2 \\
&= 2(\log \frac{1 - 2a}{a} - 1)
\end{aligned}
\tag{14}
$$

Since $0 \le a \le \frac{1}{3}$, $\frac{\mathrm{d}H(s)}{\mathrm{d}a}$ is always greater than or equal to zero. In other words, $H(s)$ increases with $a$.

However, $\frac{\mathrm{d}IG(s)}{\mathrm{d}a}$ is greater than or equal to zero when $0 \le a \le \frac{1}{4}$, and less than zero when $\frac{1}{4} < a \le \frac{1}{3}$. That being said, when $a$ is less than $\frac{1}{4}$, $IG(s)$ increases with $a$; nonetheless, when $a$ is greater than $\frac{1}{4}$, $IG(s)$ decreases with $a$, although $H(s)$ keeps increasing in this case.

To give a specific counterexample, let $a_1 = 0.25$ and $a_2 = 0.3$ and denote these nodes as $s_1$ and $s_2$ respectively. Thus, the soft labels of $s_1$ and $s_2$ are $[0.5, 0.25, 0.25]$ and $[0.4, 0.3, 0.3]$, and we have $H(s_1) = 1.5$, $H(s_2) \approx 1.571$, $IG(s_1) = 1$ and $IG(s_2) \approx 0.971$. Evidently, $H(s_1) < H(s_2)$ and $IG(s_1) > IG(s_2)$.

Therefore, the theorem follows.

## B  NORMALIZED LABEL

For a certain node, if the soft label given by the model is $[a_1, a_2, ..., a_c]$, and the oracle indicates that the node does not belong to class $l$, then the probability that the node belongs to class $i$ ($i \ne l$) is given by

$$\frac{a_i}{\sum_{j \ne l} a_j} \tag{15}$$

In essence, this is a conditional probability. Formally, let $j$ be the ground truth label of the node, and then our target can be expressed as

$$P(j = i | j \ne l) \tag{16}$$

With the definition of conditional probability, the above can be transformed as follows

$$P(j = i | j \ne l) = \frac{P(j = i, j \ne l)}{P(j \ne l)} = \frac{P(j = i)}{P(j \ne l)} \tag{17}$$

According to the known condition, $P(j = i) = a_i$ and $P(j \ne l) = \sum_{j \ne l} a_j$. Therefore, the theorem follows. The theorem suggests that, when the oracle gives a negative answer, the normalized soft label should be set with accordance to the soft label given by the model, rather than the uniform distribution.

Table 4: Overview of the Five Datasets

| Dataset | #Nodes | #Features | #Edges | #Classes | #Train/Val/Test | Task type | Description |
|---|---|---|---|---|---|---|---|
| Cora | 2,708 | 1,433 | 5,429 | 7 | 1,208/500/1,000 | Transductive | citation network |
| Citeseer | 3,327 | 3,703 | 4,732 | 6 | 1,827/500/1,000 | Transductive | citation network |
| Pubmed | 19,717 | 500 | 44,338 | 3 | 18,217/500/1,000 | Transductive | citation network |
| ogbn-arxiv | 169,343 | 128 | 1,166,243 | 40 | 90,941/29,799/48,603 | Transductive | citation network |
| Reddit | 232,965 | 602 | 11,606,919 | 41 | 155,310/23,297/54,358 | Inductive | social network |

## C  INFORMATION GAIN (IG)

For node $v_i$, the IG of annotating it is defined as

$$IG(v_i) = H(\hat{\boldsymbol{y}}_i) - P(v_i-)H(\hat{\boldsymbol{y}}_i', v_i-), \tag{18}$$

Starting from the expectation of entropy reduction, we show how it is transformed into IG as follows. Note that if the oracle agrees with the pseudo label, i.e., event $v_i+$, the hard label will be obtained, thus the entropy dropping to 0.

$$\begin{aligned}
\mathrm{E}[H(\hat{\boldsymbol{y}}_i) - H(\hat{\boldsymbol{y}}_i')] &= P(v_i+)(H(\hat{\boldsymbol{y}}_i) - H(\hat{\boldsymbol{y}}_i', v_i+)) + P(v_i-)(H(\hat{\boldsymbol{y}}_i) - H(\hat{\boldsymbol{y}}_i', v_i-)) \\
&= P(v_i+)(H(\hat{\boldsymbol{y}}_i) - 0) + P(v_i-)(H(\hat{\boldsymbol{y}}_i) - H(\hat{\boldsymbol{y}}_i', v_i-)) \\
&= (P(v_i+) + P(v_i-))H(\hat{\boldsymbol{y}}_i) - P(v_i-)H(\hat{\boldsymbol{y}}_i', v_i-) \\
&= IG(v_i)
\end{aligned} \tag{19}$$

## D  EXAMPLE OF EXPECTED INFORMATION GAIN

- *Case 1: $v_i$ belongs to the oracle annotated class $l$.* The prediction is correct in this case, and the information changes from the uncertain soft label to the one-hot label. Suppose the model predicted softmax outputs $\hat{\boldsymbol{y}}_i$ is [0.5, 0.3, 0.2], the label is changed into [1, 0, 0] and the corresponding information gain is defined as $H([0.5, 0.3, 0.2]) - H([1, 0, 0])$, where $H(\cdot)$ is the entropy function.

- *Case 2: $v_i$ doesn't belong to the oracle annotated class $l$.* In this case, the model prediction is incorrect, but the uncertainty of the soft label is also decreased because we have already known that the true label may belong to the remaining classes except for the class $l$. Similar to case 1, suppose the predicted label $\hat{\boldsymbol{y}}_i$ is [0.5, 0.3, 0.2], the soft label is changed into [0, 0.6, 0.4] and the corresponding information gain is defined as $H([0.5, 0.3, 0.2]) - H([0, 0.6, 0.4])$.

## E  TIME COMPLEXITY ANALYSIS

Given a $K$-layer GNN, the propagation matrix ($N \times N$, with $M$ nonzero values) and label matrix ($N \times c$) should be multiplicated $K$ times in IGP. Therefore, in the stage of node selection (line 3-5 in Algorithm 1), selecting the node that maximizes the propagation of information gain $F(\mathcal{V}_l \cup \{v\})$ from unlabeled nodes requires $\mathcal{O}(KMc)$, where $K$ is the number of layers of GNNs, $M$ is the number of edges and $c$ is the number of classes. All in all, the cost of selecting a batch of b nodes is $\mathcal{O}(bKMc)$. Suppose we filter out all but $n$ nodes with a degree larger than a threshold and get the corresponding propagation matrix ($n \times N$, with $m$ nonzero values), we can reduce the overall time complexity to $\mathcal{O}(bKmc)$.

## F  EXPERIMENTAL DETAILS

### F.1  DATASET DESCRIPTION

**Cora**, **Citeseer**, and **Pubmed**[1] are three popular citation network data-sets, and we follow the public training/validation/test split in GCN  Kipf & Welling (2017b). In these three networks, papers from

---

[1]https://github.com/tkipf/gcn/tree/master/gcn/data

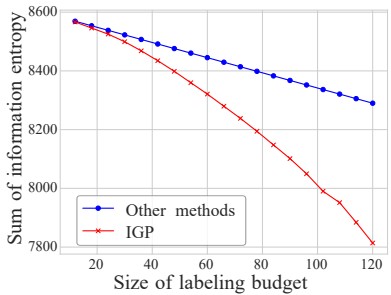

Figure 6: the change of information entropy for the whole dataset when labeling budget grows.

different topics are considered as nodes, and the edges are citations among the papers. The node attributes are binary word vectors, and class labels are the topics papers belong to.

**Reddit** is a social network dataset derived from the community structure of numerous Reddit posts. It is a well-known inductive training dataset, and the training/validation/test split in our experiment is the same as that in GraphSAGE Hamilton et al. (2017). The public version provided by GraphSAINT[2] Zeng et al. (2020) is used in our paper.

**ogbn-arxiv** is a directed graph, representing the citation network among all Computer Science (CS) arXiv papers indexed by MAG. The training/validation/test split in our experiment is the same as the public version. The public version provided by OGB[3]is used in our paper.

For more specifications about the five aforementioned datasets, see Table 4.

### F.2 IMPLEMENTATION DETAILS

For GPA Hu et al. (2020), to achieve the highest accuracy, we directly use the pre-trained model by its authors on Github. Specifically, for Cora, we choose the model pre-trained on PubMed and Citeseer; for PubMed, we choose the model pre-trained on Cora and Citeseer; for Citeseer and Reddit, we choose the model pre-trained on Cora and PubMed; for ogbn-arxiv, we choose the model pre-trained on PubMed and Reddit. Other hyper-parameters are all identical to the open-source code.

For AGE Cai et al. (2017) and ANRMAB Gao et al. (2018), to procure well-trained models and guarantee that their model-based selection criteria work well, GCN is trained for 200 epochs in each node selection iteration. AGE is implemented with its open-source version and ANRMAB in accordance with its original paper.

For ALG Zhang et al. (2021a) and GRAIN Zhang et al. (2021c), we follow the public code released by the original paper.

For the hyperparameter $\alpha$, we set it as 1 in Citeseer, PubMed and ogbn-arxiv, and as 2 for Cora and Reddit. Besides, the degree for dismissing uninfluential nodes in IGP is 8 for Cora, Citeseer, PubMed, and 15 for Reddit and ogbn-arxiv.

We carry out all the experiments on an Ubuntu 16.04 system with Intel(R) Xeon(R) CPU E5-2650 v4 @ 2.20GHz, 4 NVIDIA GeForce GTX 1080 Ti GPUs and 256 GB DRAM. All the experiments are implemented in Python 3.6 with Pytorch 1.7.1 Paszke et al. (2019) on CUDA 10.1.

### G INTERPRETABILITY ANALYSIS

We test the change of information entropy for Citeseer dataset when the labeling budget grows from $2c$ to $20c$ ($c$ is the number of classes) and report the result in Figure 6. Compare to other methods that bring fixed information entropy decrease, IGP has much less information entropy and the gap becomes larger when the labeling budget grow, which explains the better performance of our method.

---

[2]https://github.com/GraphSAINT/GraphSAINT
[3]https://ogb.stanford.edu/docs/nodeprop/#ogbn-arxiv

Table 5: Performance along with running time on the Reddit dataset.

| Method | ANRMAB | GPA | AGE | SEAL | ALG | IGP | IG | GRAIN |
|---|---|---|---|---|---|---|---|---|
| Relative Running Time | 340.00 | 331.25 | 8.25 | 4.00 | 1.63 | 1.38 | 1.25 | 1.00 |
| Test Accuracy(%) | 91.5($\pm$0.3) | 91.8($\pm$0.2) | 91.6($\pm$0.3) | 92.1($\pm$0.3) | 92.4($\pm$0.3) | **93.4($\pm$0.2)** | 91.7($\pm$0.3) | 92.5($\pm$0.1) |

Table 6: The test accuracy (%) on different hyperparameters on the PubMed dataset

| parameters | $\alpha=0$ | $\alpha=0.5$ | $\alpha=1$ | $\alpha=2$ |
|---|---|---|---|---|
| SGC | 79.6($\pm$0.7) | 82.8($\pm$0.6) | 83.2($\pm$0.6) | 82.9($\pm$0.5) |
| APPNP | 80.4($\pm$0.6) | 83.4($\pm$0.4) | 83.7($\pm$0.5) | 83.6($\pm$0.5) |
| GCN | 80.7($\pm$0.4) | 83.5($\pm$0.5) | 83.6($\pm$0.5) | 83.4($\pm$0.6) |
| MVGRL | 80.5($\pm$0.6) | 83.2($\pm$0.5) | 83.9($\pm$0.4) | 83.5($\pm$0.4) |

## H  EFFICIENCY ANALYSIS

To test the efficiency of IGP, we also compare its running time per AL iteration with other baselines in a batch of node selection in the Reddit dataset. Besides, to measure the influence of the extra computational cost in maximizing the propagation of information gain in the local neighborhood, we remove the propagation part in IGP and name this baseline IG.

We set the training time of GRAIN as the baseline, the relative training time and the test accuracy of each method are provided in Table 5. As shown in Table 5, our method IGP can get comparable efficiency as GRAIN while achieving the accuracy improvement of 0.9%. Besides, removing the propagation part in IGP will make the AL process a little faster, but it also leads to large performance degradation of 1.7% compared with IGP.

## I  EXPLORATION OF HYPERPARAMETERS

Our proposed IGP has only one hyperparameter $\alpha$, which controls the importance of the weak label. To investigate its impact on IGP, we set $\alpha$ to different values and then report the corresponding test accuracy with different base GNN models on different datasets. The experimental results in Table 6 show that removing the weak label will lead to large performance degradation in different models, which verifies the effectiveness of our proposed weak label. Besides, our method IGP is robust to the choice of $\alpha$ since its performance is stable on different models when we increase $\alpha$ from 0.5 to 2.

## J  INFLUENCE OF THE NUMBER OF CLASSES

To clarify the influence of the number of classes, we examine the model accuracies as a function of the number of classes on the ogbn-arxiv dataset. Specifically, we gradually increase the number of classes and the corresponding number of nodes of the ogbn-arxiv dataset, and report the mean test accuracy of 10 runs with and without the soft-label. As shown in the newly added Figure 7(a), although more classes increase the complexity of a classification problem (thus a decreasing model accuracy), we see that the performance gain of adding soft labels decreases slightly as the number of classes increases.

The reason is that although the soft-max is taken over the remaining classes, the distribution of the output of soft-max (i.e., soft label) is expected to be highly skewed, which can help to alleviate the effect of increased class number on the quality of the soft label. To demonstrate this, we add a new Figure 7(b) showing the percentage of the node whose top-K accuracy is larger than 80%. As the percentages of top K class increases (i.e., K/total class number), more than 90% nodes only need the predicted top 20% classes to guarantee the top-K accuracy larger than 80%, which means few classes predominate the softmax distribution.

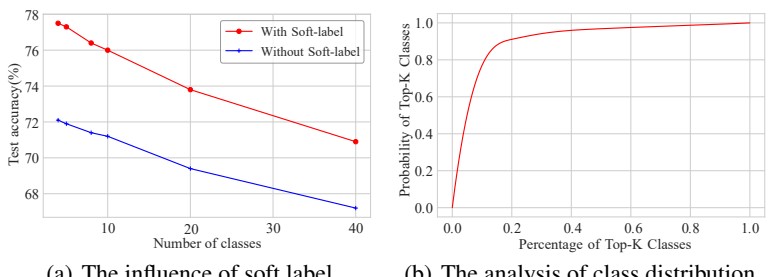

(a) The influence of soft label.    (b) The analysis of class distribution.

Figure 7: (Left) The test accuracy of the different number of classes with and without the soft label. (Right) The relationship of probability and percentage of top-K class in all of the classes.

Table 7: The test accuracy (%) with different filtering degrees on the Reddit dataset.

| Filtering Degree | 0 | 5 | 10 | 15 |
|---|---|---|---|---|
| Test Accuracy (%) | 93.6($\pm$0.3) | 93.5($\pm$0.3) | 93.4($\pm$0.2) | 93.4($\pm$0.2) |

## K  ANALYSIS OF SAMPLE COMPLEXITY

With a non-perfect oracle, our method could reduce sample complexity compared with other AL strategies because we improve the utilization of queries and samples. Traditional AL algorithms have to discard the data samples of queries that have not obtained hard labels from non-perfect Oracle. By contrast, we could still leverage these data samples in a soft-label manner, thus reducing the error quickly than other AL algorithms.

Although some AL algorithms enjoy theoretical guarantees on the traditional linear machine learning models (e.g., linear regression), analyzing the sample complexity of AL for deep learning is quite difficult due to the non-linear nature of neural networks. Most sample complexity is analyzed under the passive learning for neural networks in the literature. In addition to the non-linear nature of neural networks, the graph neural networks (GNNs) are even more complicated as graph data is no longer independent and identically distributed (iid), making the theoretical analysis of sample complexity extremely difficult for GNN-based AL.

## L  ANALYSIS OF THE FILTERING DEGREE

To clarify how much accuracy would drop when we filter the small-degree nodes, we increase the filtering degree from 0 to 15 and then report the corresponding mean test accuracy of 10 runs with GCN in Table 7. Note that the threshold we used in the previous experiments is 15 in Reddit.

As shown in Table 7, the test accuracy only drops slightly by 0.2% when we increase the threshold from 0 to 15, meaning that our efficiency optimization will not introduce a large negative influence on the prediction ability of the GCN model. Compared with the uninfluential nodes with small node degrees, labeled nodes will larger degrees will propagate their label information to more adjacent nodes and thus make larger entropy reduction for the whole dataset. As a result, nodes with larger degrees have a larger probability of being selected in IGP even if the efficiency optimization is not introduced (i.e., setting the filtering degree to 0).

