# OpenReview forum: "Information Gain Propagation: a New Way to Graph Active Learning with Soft Labels"
_ICLR.cc/2022/Conference — ICLR 2022 Poster_

### Official Review · Reviewer_tKPB · 2021-10-26

**Correctness:** 3
**Technical Novelty And Significance:** 2
**Empirical Novelty And Significance:** 2
**Recommendation:** 1
**Confidence:** 4

**Main Review:**

Strong points:
1.	The paper provides a relative new framework for GNNs where the information gain propagation is maximized.
2.	The setting of relaxed queries seems to be facilitative in this case.
3.	Results show some advantage over other methods
Weak points:
1.	Motivation: except for the motivation for the relaxed queries, I do not see much motivation and discussion in the paper, especially a discussion around the state of the art in this area and how it related to the proposed method. Based on section 4.4 is the only advantage of IGP over others is the use of relaxed queries? What about the query criteria itself? What are its advantages relative to prior works.
There are many other mssign motivation throught the paper for example after the IG definition why is this a good a good tool to use?
Why relaxed queries are important, where in real life could this make a difference?
2.	Paper organization: is section 2.3 meant to be related work? How is it related to section 4.4 on prior work?
3.	Technical errors:
a.	equation (8) isn’t that supposed to be v_i – on the last component?
b.	Equation (4) the indicator is on I – an index of a node equal l – a class? That is incorrect use of the two different symbols in the indicator function
c.	With respect to what is the entropy in eq (7) computed? What are the probabilities?
4.	Theoretical foundations: there is no sample complexity analysis or any optimality analysis for the given criterion. The only theorem provided is a rather trivial one showing that entropy doesn’t coincide with IG selection criterion.
5.	Clarity:
a.	Unclear sentences like: “Since the influence magnitude is diverse to the influenced nodes…” what does it mean?
6.	Experimental validation: I am left unimpressed by the rate of improvement shown by the proposed algorithm. Moreover, the analysis lacks error bars and standard deviation for the 10 executions.


**Summary Of The Paper:**

The paper proposes an active learning method for GNNs that is based on an information gain maximization where ethe information gain is obtained by querying a data point and looking at the influence of the queried node on the neighborhood relative to their previous information.  They also claim the setting of relaxing the oracle answer to be a binary confirmation of the most probable label. Experiments are presented where some advantage is shown for the method.

**Summary Of The Review:**

Strong points:
1.	The paper provides a relative new framework for GNNs where the information gain propagation is maximized.
2.	The setting of relaxed queries seems to be facilitative in this case.
3.	Results show some advantage over other methods
Weak points:
1.	Motivation: except for the motivation for the relaxed queries, I do not see much motivation and discussion in the paper, especially a discussion around the state of the art in this area and how it related to the proposed method. Based on section 4.4 is the only advantage of IGP over others is the use of relaxed queries? What about the query criteria itself? What are its advantages relative to prior works.
There is lacking motivation throughoutt the paper for example after the IG definition why is this a good a good tool to use?
Why relaxed queries are important, where in real life could this make a difference?
2.	Paper organization: is section 2.3 meant to be related work? How is it related to section 4.4 on prior work?
3.	Technical errors:
a.	equation (8) isn’t that supposed to be v_i – on the last component?
b.	Equation (4) the indicator is on I – an index of a node equal l – a class? That is incorrect use of the two different symbols in the indicator function
c.	With respect to what is the entropy in eq (7) computed? What are the probabilities?
4.	Theoretical foundations: there is no sample complexity analysis or any optimality analysis for the given criterion. The only theorem provided is a rather trivial one showing that entropy doesn’t coincide with IG selection criterion.
5.	Clarity:
a.	Unclear sentences like: “Since the influence magnitude is diverse to the influenced nodes…” what does it mean?
6.	Experimental validation: I am left unimpressed by the rate of improvement shown by the proposed algorithm. Moreover, the analysis lacks error bars and standard deviation for the 10 executions.
7.	There is not discussion of parameter tuning nor running time of the proposed method.

---

> ### Author Response · Authors · 2021-11-17
> **Response to Reviewer-4, Part 1**
>
> Thank you for your review and valuable feedback.
>
> ### 1. Motivation: advantages of IGP
>
> **(1)Relaxed query**
> The reviewer raises a good question of “why relaxed queries are important and where could this make a difference” in real life. Our motivation here is that it is too strong to require oracles to label instances that may be out of their domain knowledge. The importance of our new form of queries lies in the relaxation of the traditional assumption, extending AL to the applications where oracles are not perfect and have knowledge blind spot. The difference we made here is that we allow an oracle to admit that he is incapable of labeling some query instances and simply answer “this data does not belong to the domain (label) I am familiar with.”
>
> **(2)Query criteria**
> Another important contribution of this paper is the new query criteria itself under the relaxed query. Its advantages relative to prior works lie in the following two points. First, we propose the new IG criteria that further exploits soft labels in relaxed queries and their supervision in training. Second, we extend IG to IGP that explicitly employs the influence propagation of GNNs to achieve much higher performance. This is the reason why our method is a good tool to use for AL on GNNs.
>
> **(3)Discussion around state of the art**
> Section 2.3 and Section 4.4 discuss the related AL methods. The key difference here is that existing SOTA methods still fall into the routine of perfect Oracle that is capable of providing a hard label for each query. By contrast, IGP proposes a new AL research problem and strategy under relaxed query (e.g., Oracle with knowledge blind spot).
>
>
>
> ### 2. Paper organization
> The reviewer is right that both Sec 2.3 and Sec. 4.4 are related work, which focuses on traditional AL methods for general models and GNN models, respectively. As suggested by the other reviewer, we combine them together.
>
> ### 3. Notation
> We thank the reviewer for pointing out the possible ambiguity of using the same indicator notation $i$ in Equation (4), we replace $i$ with another indicator notation $t$ to avoid such ambiguity. Besides, the last component in Equation (8) is $v_i-$.In Equation (7), $I_f(v_j,v_m,k)$ can be computed by Equation (6) and $\hat{\boldsymbol{y}}$ is the model predicted probabilities produced by the GNN models.
>
> ### 4. Theoretical foundations
> **(1)complexity analysis**
> Thank you for bringing this to our attention. We add the complexity analysis of IGP in Appendix. E. Given a $K$-layer GNN, the propagation matrix ($N\times N$, with $M$ nonzero values) and label matrix ($N \times c$) should be multiplicated $K$ times in IGP. Therefore, in the stage of node selection (line 3-5 in Algorithm 1), selecting the node that maximizes the propagation of information gain $F(\mathcal{V}_l \cup \{v\})$ from unlabeled nodes requires $\mathcal{O}(KMc)$, where $K$ is the number of layers of GNNs, $M$ is the number of edges, and $c$ is the number of classes. In all, the cost of selecting a batch of b nodes is $\mathcal{O}(bKMc)$.
>
>
>
> **(2)The importance of theorem**
> We would like to highlight the importance of the theorem, which points out that the general uncertain measure (entropy) under traditional AL query is not optimal under our new AL setting with relaxed query. This provides a theoretical interpretation for the requirement of new IG criteria.
>
> ### 5. Clarity
> We clarify the sentence as the influence of a node on its different neighbors can be various. For example, compared with the distant neighbors, a labeled node is more likely to have larger influence on its adjacent neighborhood nodes.”

---

> > ### Author Response · Authors · 2021-11-17
> > **Response to Reviewer-4, Part 2**
> >
> > ### 6. Experimental validation
> > We add the standard deviation for Table 1, Table 2, and Table 3. It is worth to point our method improves SOTA by up to 2.2%, which is a significant accuracy improvement.
> >
> > ### 7. parameter tuning & running time
> > We add the experiment on the runtime of different methods for each batch selection in Table 5, showing that our method achieve highest accuracy and good efficiency. Here we measure the cost by the relative selection time to that of GRAIN approach.
> >
> > **Table 5: Performance along with running time on the Reddit dataset.**
> >
> > |        Methods        |  ANRMAB  |   GPA    |   AGE    |   SEAL   |   ALG    |   IGP    |    IG    |  GRAIN   |
> > |:---------------------:|:--------:|:--------:|:--------:|:--------:|:--------:|:--------:|:--------:|:--------:|
> > | Relative Running Time |  340.00  |  331.25  |   8.25   |   4.00   |   1.63   |   1.38   |   1.25   |   1.00   |
> > |   Test Accuracy(\%)    | 91.5±(0.3) | 91.8±(0.2) | 91.6±(0.3) | 92.1±(0.3) | 92.4±(0.3) | **93.4±(0.2)** | 91.7±(0.3) | 92.5±(0.1) |
> >
> >
> > We also add the discussion of parameter tuning. Our proposed IGP has only one hyperparameter $\alpha$, which controls the importance of a weak label. To investigate its impact on IGP, we set $\alpha$ to different values and then report the corresponding test accuracy with different base GNN models on different datasets. The experimental results in Table 6 show that removing the weak label will lead to large performance degradation in different models, which verifies the effectiveness of our proposed weak label. Besides, our method IGP is robust to the choice of $\alpha$ since its performance is stable on different models when we increase $\alpha$ from 0.5 to 2.
> >
> > **Tabel 6: The test accuracy (\%) on different hyperparameters on the PubMed dataset.**
> >
> > | parameters | $\alpha$=0 | $\alpha$=0.5 | $\alpha$=1 | $\alpha$=2 |
> > |:----------:|:----------:|:------------:|:----------:|:----------:|
> > |    SGC     | 79.6(±0.7) |  82.8(±0.6)  | 83.2(±0.6) | 82.9(±0.5) |
> > |   APPNP    | 80.4(±0.6) |  83.4(±0.4)  | 83.7(±0.5) | 83.6(±0.5) |
> > |    GCN     | 80.7(±0.4) |  83.5(±0.5)  | 83.6(±0.5) | 83.4(±0.6) |
> > |   MVGRL    | 80.5(±0.6) |  83.2(±0.5)  | 83.9(±0.4) | 83.5(±0.4) |

---

> > > ### Comment · Reviewer_tKPB · 2021-11-18
> > > **Thank you for providing these results**
> > >
> > > Thank you for the running time results which are encouraging, and the parameter sensitivity experiment as well. As for the accuracy difference, 2.2% is convincing if the standard deviation is showing it is a significant difference. Since you didnt report that so far it is hard to judge.

---

> > > > ### Author Response · Authors · 2021-11-19
> > > > **Response to the standard deviation**
> > > >
> > > > We have updated the standard deviation in all the tables (Table 1,2,3,5,6). Specifically, we show Table 1 as follows and others in the revised manuscript.
> > > >
> > > > **Table 1: The test accuracy (%) on different datasets with the same labeling budget.**
> > > >
> > > > | Methods |      Cora       |     Citeseer      |      PubMed       |     Reddit      |     ogbn-arxiv      |
> > > > |:-------:|:--------------:|:--------------:|:--------------:|:--------------:|:--------------:|
> > > > Random|78.8(±0.8)|70.8(±0.9)|78.9(±0.6)|91.1(±0.5)|68.2(±0.4)|
> > > > AGE|82.5(±0.6)|71.4(±0.6)|79.4(±0.4)|91.6(±0.3)|68.9(±0.3)|
> > > > ANRMAB|82.4(±0.5)|70.6(±0.6)|78.2(±0.3)|91.5(±0.3)|68.7(±0.2)|
> > > > | GPA     | 82.8(±0.4)    | 71.6(±0.4)     | 79.9(±0.5)     | 91.8(±0.2)     | 69.2(±0.3)     |
> > > > | SEAL    | 83.2(±0.5)    | 72.1(±0.4)     | 80.3(±0.4)     | 92.1(±0.3)     | 69.5(±0.1)     |
> > > > | ALG     | 83.6(±0.6)    | 73.6(±0.5)     | 80.9(±0.3)     | 92.4(±0.3)     | 70.1(±0.2)     |
> > > > | GRAIN   | 84.2(±0.3)    | 74.2(±0.3)     | 81.8(±0.2)     | 92.5(±0.1)     | 70.3(±0.2)     |
> > > > | **IGP** | **86.4±0.6)** | **75.8(±0.3)** | **83.6(±0.5)** | **93.4(±0.2)** | **70.9(±0.3)** |

---

> > ### Comment · Reviewer_tKPB · 2021-11-18
> > **Sample Complexity, motivation, technical errors**
> >
> > Sample complexity: In my review I was referring to a missing classic sample complexity analysis for active learning. Namely, how many samples are needed to be queried by IGP to reduce the error by a \delta. The authors instead provide in their response a running time complexity, which was also missing in their paper but is not related to sample complexity.
> >
> > Technical errors: you still didnt clarify regarding my item 6
> >
> > Relaxed queries: does active learning become more efficient with relaxed queries, i.e., does the number of queries lower when compared with using a perfect oracle. what is really the effect of relaxing the problem? none of these is thoroughly discussed.

---

> > > ### Author Response · Authors · 2021-11-19
> > > **Response to technical errors, relaxed queries, and sample complexity**
> > >
> > > We thank the reviewer for the continuous comments.
> > >
> > > ### 1. Technical errors
> > > We have corrected the notation errors pointed out by the reviewer in Equation (4) and Equation (8) as follows:
> > > Equation(4)
> > >
> > > \begin{equation}
> > > \small
> > >     \begin{aligned}
> > >     a_t\^{\prime}=
> > >     \begin{cases}
> > >     \mathbb{1}(t = l) \& v_i\~\text{belongs to class $l$}\\\\
> > >     \frac{a_t\cdot \mathbb{1}(t = l)}{\sum_{j \neq l}a_j} \& v_i~\text{does not belong to class $l$}
> > >     \end{cases}
> > >     \end{aligned}
> > > \end{equation}
> > >
> > > Equation(8)
> > >
> > > \begin{equation}
> > > \begin{aligned}
> > >     IGP(v_j,v_i,k)= P(v_i+)IGP(v_j,v_i,k,v_i+) + P(v_i-)IGP(v_j,v_i,k,v_i-)
> > > \end{aligned}
> > > \end{equation}
> > >
> > > For Eq.(7), we compute the marginal gain of entropy reduction on a specific unlabeled node $v_j$ if we select a node $v_i$ to query from the oracle. In particular, for a specific node $v_j$, each labeled node $v_m$ $\in V_l$ would have a different label-smoothing effect on node $v_j$, captured by the influence magnitude function $I_f(v_j,v_m, k)$ in Eq.(6). So the aggregated smoothed label for node $v_j$ is given by $\sum_{v_m\in \mathcal{V}_l}I_f(v_j,v_m,k)\hat{\boldsymbol{y}}_m^{\prime}$. Notice here $\hat{\boldsymbol{y}}_m^{\prime}$  is the normalized soft label defined in Eq.(4), which is a probability distribution over the possible classes. Therefore, the entropy in Eq.(7) actually computes the entropy of aggregated smoothed label for node $v_j$. The probabilities here indicate which class the node $v_j$ belongs to based on the label-smoothing effect in GNNs.
> > >
> > > ### 2. Relaxed queries
> > > We clarify here that we are not arguing that active learning becomes more efficient with relaxed queries than those using a perfect oracle. Instead, we argue that it is too strong in traditional AL to assume that oracles may always behave perfectly, requiring all oracles to label instances that may be out of their domain knowledge.
> > >
> > > Therefore, this paper handles a different yet more challenging AL setting where the assumption of a perfect oracle might not be met. We allow an oracle to admit that he/she is incapable of labeling some query instances and simply answer “this data does not belong to my familiar domain (label)”. The effect of such relaxation on traditional AL algorithms is that they cannot leverage queries that fail to obtain a hard label from the non-perfect oracle. By contrast, our method could still leverage these queries based on the soft label. In particular, we exclude the label disagreed by the oracle from the model prediction and generate a soft-max (soft label) taken over the remaining classes to provide supervision with our defined objective function in Equation (10). Therefore, compared to traditional AL algorithms, we improve the utilization of relaxed queries under non-perfect Oracle.
> > >
> > >
> > >
> > > ### 3. Sample complexity.
> > > Thank you for bringing sample complexity to our attention. With a non-perfect oracle, our method could reduce sample complexity compared with other AL strategies because we improve the utilization of queries and samples. As we explained earlier, traditional AL algorithms have to discard the data samples of queries that have not obtained hard labels from non-perfect Oracle. By contrast, we could still leverage these data samples in a soft-label manner, thus reducing the error quickly than other AL algorithms.
> > >
> > > Although some AL algorithms enjoy theoretical guarantees on the traditional linear machine learning models (e.g., linear regression), analyzing the sample complexity of AL for deep learning is still an open research challenge due to the non-linear nature of neural networks. Most sample complexity is analyzed under the passive learning for neural networks in the literature. To the best of our knowledge, only another submission [1] in ICLR 2022 studies the sample complexity problem for training neural networks, but the analysis is limited to the simplest one-hidden layer neural networks. In addition to the non-linear nature of neural networks, the graph neural networks (GNNs) are even more challenging as graph data is no longer independent and identically distributed (iid). We have added the above discussion in Appendix.K in the revised manuscript.
> > >
> > > [1] Zhao Song, Baocheng Sun, Danyang Zhuo.  [*Sample Complexity of Deep Active Learning.*](https://openreview.net/forum?id=PU3VGS93gxD) Under review at ICLR 2022.

---

### Official Review · Reviewer_QqqC · 2021-10-31

**Correctness:** 4
**Technical Novelty And Significance:** 4
**Empirical Novelty And Significance:** 3
**Recommendation:** 8
**Confidence:** 5

**Details Of Ethics Concerns:**

I do NOT find so-said concerns.

**Main Review:**

1.  The strengths:
1.1. present using the relaxied queries to replace commonly-used exact labeling strategy for AL, i.e., only judging the correctness of the predicted labels (a binary question) rather than identifying the exact class (a multi-class question), doing so is relatively easier and to be my best knowledge, it seems hardly to be done before.
1.2. provide a new criteria for active learner with the help of such relaxed queries and soft labels via maximizing defined the IGP.
1.3. obtain significant performance boost comparing with SOTAs.
One the whole, the proposed method can have more flexibility, e.g., being applicable to a large variety of GNN variants.
2. The  weaknesses:
2.1 Class-imbalance problem;
2.2. Due to GNN characteristics, IGP inherits possible oversmoothness problem.
In addition, please concern possible relation with "Two-dimensional active learning for image classification" availavle at internet.

**Summary Of The Paper:**

The submissiom presents a GNN-based AL method using soft-label technique with 2 innovations different from existing works
1. using relaxed queries and 2. building a new criteria of maximizing IGP for active learner with relaxed queries and soft labels. 3.outperforming SOTAs in performance.

**Summary Of The Review:**

In this submission, in my opinion, the authors consider using relaxied queries to replace exact labeling for active learner and define a new criterion with IGP  for learning. Using the relaxied queries indeed provides the oracle a relatively easier labeling andthus more reliable. While Theorem 3.1 gives some insight. The authors provide their codes for REPRODUCIBILITY and the experiments are also relatively sufficiently conducted and compared, the results are convincing!

---

> ### Author Response · Authors · 2021-11-17
> **Response to Reviewer-3**
>
> Thanks for your positive review of our submission and valuable feedback.
> ### 1. Class-imbalance Problem
> Thank you for bringing this to our attention. We agree with the reviewer that the class-imbalance problem has not been considered here since it is not the main focus of this paper. However, this is a promising direction to move our approach forward through explicitly introducing class-balancing constraints in the query process to reduce the imbalance of the labeled subset. For example, as a means of balancing, priority is given to nodes whose predicted classes are underrepresented among those already labeled. We will add the discussion in the paper.
>
> ### 2. Over-smoothness problem
> Our method focuses on the query process to provide labels for GNNs, which are orthogonal to and compatible with the over-smoothing combatting technique. For example, as insightfully pointed out by the reviewer, our method is applicable to many GNN variants (as shown in Table). Thus, one could select those that prevent over-smoothing, such as the APPNP model.
>
> ### 3. Paper Relation
> Thanks for pointing out to us the related Two-Dimensional Active Learning (2DAL) work. 2DAL is a two-dimensional active learning strategy, which selects the most “informative” sample-label pairs to reduce the uncertainty along the dimensionalities of both sample and label. The key difference here is that 2DAL leverages the label correlation under a multi-label setting. By contrast, we leverage the labels correlation of the connected samples given graph structure, which is still along the sample dimension under a single-label setting. We have added the discussion of 2DAL in the paper, which inspires us further to adapt AL to graphs under a multi-label setting.

---

### Official Review · Reviewer_owkR · 2021-11-02

**Correctness:** 3
**Technical Novelty And Significance:** 3
**Empirical Novelty And Significance:** Not applicable
**Recommendation:** 5
**Confidence:** 4

**Main Review:**

Strengths:
1. The relaxed query setting is well motivated for AL and it seems to be applicable in real-life scenarios.
2. The proposed AL criterion takes both information gain and influence magnitude into consideration, which can explicitly maximise the propagation of information gain.
3. The source code of the paper is released to ensure the reproductivity of results.
4. Overall, this paper is well-written, though there are some typos in equations and text.

Weakness:
1. No complexity analysis on the AL query criterion of IGP. As IGP needs to maximise the propagation of information gain in local neighbourhood, it is expected to incur extra computational overhead.
2. Missing discussion and comparison with recently published GNN-based AL methods, e.g.,
Li et al. SEAL: Semi-supervised adversarial active learning on attributed graphs, IEEE TNNLS, 32 (7), 3136-3147, 2021.

3. The discussions on related work are respectively given in Section 2.3 and Section 4.4. Would it be better to combine the two sections?
4. Some symbols used in Eq.(8) are incorrect. e.g., would it be v_i - in the last term?


**Summary Of The Paper:**

This paper proposes a new GNN-based active learning (AL) method for graph data, under a relaxed query setting where the oracle can only judge the correctness of the predicted labels. A new AL query criterion is proposed to select the nodes that can maximise the information gain propagation (IGP) in local neighbourhood.


**Summary Of The Review:**

This paper considers a new AL setting with relaxed queries. Under this setting, a new AL query criterion is proposed to incorporate soft labels and information gain propagation. The setting and idea of this paper are interesting, but there are some concerns, e.g,, the lack of computational analysis of the AL query criterion and comparison to some state-of-the-art GNN-based AL methods.

---

> ### Author Response · Authors · 2021-11-17
> **Response to Reviewer-2,  Part 1**
>
> Thanks for your insightful feedback. We appreciate your assessment about this paper being "well motivated for AL and it seems to be applicable in real-life scenarios". The answers to your concerns are as follows.
>
> ### 1. Computational Complexity Analysis
>
> Thank you for bringing this to our attention. We add the complexity analysis of IGP in Appendix. E.
> Given a $K$-layer GNN, the propagation matrix ($N\times N$, with $M$ nonzero values) and label matrix ($N \times c$) should be multiplicated $K$ times in IGP.
> Therefore, in the stage of node selection (line 3-5 in Algorithm 1), selecting the node that maximizes the propagation of information gain $F(\mathcal{V}_l \cup \{v\})$ from unlabeled nodes requires $\mathcal{O}(KMc)$, where $K$ is the number of layers of GNNs, $M$ is the number of edges and $c$ is the number of classes.
> For efficiency optimization, we only take $n$ nodes with degree larger than a threshold as candidates, and get the corresponding propagation matrix ($n\times N$, with $m$ nonzero values), we can reduce the overall time complexity to $\mathcal{O}(bKmc)$.
> To effectively mitigate the cost, we could use the degree to identify and dismiss uninfluential nodes. Table 5 shows the cost of IGP selection for each batch, where we filter out the nodes with degrees smaller than 15 in Reddit. Here we measure the cost by the relative selection time to that of GRAIN approach. To measure the extra cost introduced by the propagation, we also add a baseline that only use IG without propagation. From the table we see that our method IGP can get comparable efficiency as GRAIN, while achieving the accuracy improvement of 0.9\%. Compared to IG, introducing propagation IGP can improve 1.7\% accuracy while only increasing the cost slightly.
>
> **Table 5: Performance along with running time on the Reddit dataset.**
>
> |        Methods        |  ANRMAB  |   GPA    |   AGE    |   SEAL   |   ALG    |   IGP    |    IG    |  GRAIN   |
> |:---------------------:|:--------:|:--------:|:--------:|:--------:|:--------:|:--------:|:--------:|:--------:|
> | Relative Running Time |  340.00  |  331.25  |   8.25   |   4.00   |   1.63   |   1.38   |   1.25   |   1.00   |
> |   Test Accuracy(\%)    | 91.5±(0.3) | 91.8±(0.2) | 91.6±(0.3) | 92.1±(0.3) | 92.4±(0.3) | **93.4±(0.2)** | 91.7±(0.3) | 92.5±(0.1) |

---

> > ### Author Response · Authors · 2021-11-17
> > **Response to Reviewer-2, Part 2**
> >
> > ### 2. Comparison to SEAL
> >
> > Thanks for pointing out this related work. We have added the comparison with SEAL, with the result shown in Table 1 and Table 2. SEAL devises a novel AL query strategy for node classification in an adversarial way: the divergence score generated by the discriminator serves as the informativeness measure to select the most informative node to be labeled by an oracle. From the Table 2, the experimental results show that SEAL could outperform AL algorithms using the general uncertainty measure, such as AGE, ANRMAB, and GPA.
> >
> > However, SEAL still falls into the routine of perfect Oracle that is capable of providing hard label for each query node. By contrast, IGP is a new AL query strategy with relaxed query (e.g., Oracle with knowledge blind spot), its novelty lies in explicitly exploiting the (1) soft labels for extra supervision, and (2) influence propagation over graph structure to enhance semi-supervision. As a result, we find that IGP outperforms SEAL.
> >
> > **Tabel 1: The test accuracy (\%) on different datasets with the same labeling budget.**
> >
> > | Methods |      Cora       |     Citeseer      |      PubMed       |     Reddit      |     ogbn-arxiv      |
> > |:-------:|:--------------:|:--------------:|:--------------:|:--------------:|:--------------:|
> > Random|78.8(±0.8)|70.8(±0.9)|78.9(±0.6)|91.1(±0.5)|68.2(±0.4)|
> > AGE|82.5(±0.6)|71.4(±0.6)|79.4(±0.4)|91.6(±0.3)|68.9(±0.3)|
> > ANRMAB|82.4(±0.5)|70.6(±0.6)|78.2(±0.3)|91.5(±0.3)|68.7(±0.2)|
> > | GPA     | 82.8(±0.4)    | 71.6(±0.4)     | 79.9(±0.5)     | 91.8(±0.2)     | 69.2(±0.3)     |
> > | SEAL    | 83.2(±0.5)    | 72.1(±0.4)     | 80.3(±0.4)     | 92.1(±0.3)     | 69.5(±0.1)     |
> > | ALG     | 83.6(±0.6)    | 73.6(±0.5)     | 80.9(±0.3)     | 92.4(±0.3)     | 70.1(±0.2)     |
> > | GRAIN   | 84.2(±0.3)    | 74.2(±0.3)     | 81.8(±0.2)     | 92.5(±0.1)     | 70.3(±0.2)     |
> > | **IGP** | **86.4±0.6)** | **75.8(±0.3)** | **83.6(±0.5)** | **93.4(±0.2)** | **70.9(±0.3)** |
> >
> > **Table 2: Test accuracy of different models on PubMed**
> >
> > | Methods |      SGC       |     APPNP      |      GCN       |     MVGRL      |
> > |:-------:|:--------------:|:--------------:|:--------------:|:--------------:|
> > | Random  |   77.6(±0.8)   |   79.2(±0.6)   |   78.9(±0.6)   |   79.3(±0.5)   |
> > |   AGE   |   78.8(±0.5)   |   79.9(±0.5)   |   79.4(±0.4)   |   79.9(±0.4)   |
> > | ANRMAB  |   77.8(±0.4)   |   78.7(±0.5)   |   78.2(±0.3)   |   78.9(±0.3)   |
> > |   GPA   |   79.5(±0.6)   |   80.2(±0.4)   |   79.9(±0.5)   |   80.4(±0.3)   |
> > |  SEAL   |   79.8(±0.5)   |   80.5(±0.5)   |   80.3(±0.4)   |   80.7(±0.3)   |
> > |   ALG   |   80.5(±0.4)   |   81.2(±0.5)   |   80.9(±0.3)   |   81.3(±0.2)   |
> > |  GRAIN  |   81.1(±0.3)   |   82.0(±0.4)   |   81.8(±0.2)   |   82.1(±0.1)   |
> > | **IGP** | **83.2(±0.6)** | **83.7(±0.5)** | **83.6(±0.5)** | **83.9(±0.4)** |
> >
> >
> > ### 3. Related work
> > Thanks for your helpful suggestion. we combine Section 2.3 and Section 4.4 into one related work section.
> >
> > ### 4. Typo
> > Thanks for pointing us this typo. We have corrected it in the revision.

---

> > > ### Comment · Reviewer_owkR · 2021-11-21
> > > **Thanks for providing more comparison results.**
> > >
> > > Thank you for adding new comparisons with the recently proposed method (SEAL). The results are more convincing and encouraging.

---

> > ### Comment · Reviewer_owkR · 2021-11-21
> > **Thanks for providing running time comparisons**
> >
> > Thanks for adding running time results for AL queries. IGP appears to be efficient on sparse datasets where the number of edges has a similar magnitude of the number of nodes. When you filter out small-degree nodes on Reddit, could you please clarify how much accuracy would drop as a result?

---

> > > ### Author Response · Authors · 2021-11-21
> > > **Analysis of filtering degree**
> > >
> > > Thanks for the your continuous comments. To clarify how much accuracy would drop when we filter the small-degree nodes, we increase the filtering degree from 0 to 15 and then report the corresponding mean test accuracy of 10 runs with GCN in Table 7. Note that the threshold we used in the previous experiments is **15 in Reddit**.
> > >
> > >
> > > **Table 7: The test accuracy (%) with different filtering degrees on the Reddit dataset.**
> > >
> > > | Filtering Degree|0|5|10|15|
> > > |:-:|:-:|:-:|:-:|:-:|
> > > |Test Accuracy (%)|93.6(±0.3)| 93.5(±0.3)| 93.4(±0.2)| 93.4(±0.2)|
> > >
> > > As shown in Table 7, the test accuracy only drops slightly by 0.2\% when we increase the threshold from 0 to 15, meaning that our efficiency optimization will not introduce a large negative influence on the prediction ability of the GCN model.
> > >
> > > We have added the above analysis in Appendix. L in the revised manuscript.
> > >
> > > We are very glad to respond if you have any new questions.
> > >
> > > Respectfully,
> > >
> > > Paper607 Authors

---

### Official Review · Reviewer_5Qmn · 2021-11-02

**Correctness:** 3
**Technical Novelty And Significance:** 3
**Empirical Novelty And Significance:** 3
**Recommendation:** 6
**Confidence:** 4

**Main Review:**

Originality & Quality: The paper is well written and puts itself nicely in context of previous work. The overall presentation of the paper is good except a couple typos. The proposed approach is a novel adaptation of AL to graphs (to the best of my knowledge).



1). The idea of soft-labeling is intriguing! Though, the example that the paper provides in Section 1 regarding arXiv subject categories is misleading. The paper argues that a node may belong to more than one categories, e.g., mathematical finance and computational finance and that is hard for a human labeler to classify. However, it is unclear how the proposed approach solves that problem since the approach still relies on a model to make the label prediction, e.g., mathematical finance in the case above and then the human accepts it. So, we still have the same problem that the example was provided to get rid of. This is so because the example of arXiv classification is more of a multi-label classification problem and hence not a good example to motivate this paper in my opinion.



2). It will be interesting to see the model accuracies as a function of the number of classes. If the humans doesn't agree with the model, the label is discarded and a soft-max is taken over the remaining classes. So it will be interesting to see how this weak labeling by the remaining classes performs when the number of classes is large and hence the prediction problem is hard.



Significance: The paper addresses an important problem of active learning on graphs using soft-labels.

**Summary Of The Paper:**

The paper proposes a new method active learning (AL) on graphs. Unlike other AL approaches, the proposed approach provides soft labels via *relaxed queries* to the *domain experts*.



Main Contributions:

1). The paper proposes a new innovative approach for graph active learning with soft labels. The key idea is to ask a human regarding whether the model prediction is correct or not (a binary classification task) as opposed to asking them the correct "hard label" of the node. The incorrect model predictions are also not "thrown away" and used as indirect supervision by performing a soft-max over the remaining classes. This leads to a new criteria for active learning, called "maximizing information gain propagation" as opposed to maximizing entropy as done by standard AL.


2). Results are shown on a variety of real-world datasets which show the superior performance of the proposed method in achieving higher test accuracy with a certain labeling budget.

**Summary Of The Review:**

The paper provides an intriguing idea of soft-labeling to improve active learning (AL) on graphs. It seems more efficient than the conventional AL approaches on graphs and is more cost efficient. The results are mostly strong, except it is unclear what is the impact of the number of classes on the output performance.

---

> ### Author Response · Authors · 2021-11-17
> **Response to Reviewer-1 5Qmn**
>
> We appreciate your assessment about this paper being "a novel adaptation of AL to graphs ". Thanks for your constructive feedback! We believe that addressing this feedback will make our paper stronger.
>
> ### 1. Motivation & Example
>
> Thank you for bringing this to our attention. We clarify that we are not arguing that a node may belong to more than one category. Instead, we argue that it is too strong in traditional AL to assume that an oracle is capable of providing labeling information for each query instance. In our example, the task in ogbn-papers100M is to leverage the citation graph to infer the labels of the arXiv papers into 172 arXiv subject areas (a single-label, 172-class classification problem). In this example, a specialist/expert in the subject areas of machine learning is incapable of labeling query instances with subject areas of finance (such as mathematical finance or computational finance), which is out of his domain knowledge.
>
> To relax the traditional assumption, we define a new AL query strategy for Oracle with knowledge blind spot (KBS), which allows an oracle to admit that he/she is incapable of labeling some query instances and simply answer “this data does not belong to my familiar domain (label)”. Under this new strategy, we define new IG criteria to obtain the largest uncertain reduction even when the unlabeled instance belongs to the oracle’s KBS. The key insight here is that each query process is useful from the perspective of information gain, and answering “this data does not belong to the certain class” also brings entropy reduction, hence supervision. Inspired by this insight, we propose the new IG criteria considering that the oracle will "agree/disagree" with the pseudo label. If the Oracle disagrees with the model, the label is discarded, and a soft-max (soft label) is taken over the remaining classes. The updated soft label still provides weak supervision with our defined objective function in Equation (10).
>
> By contrast, in the traditional AL approach requiring a hard label, if the oracle doesn't agree with the model, the label has to be discarded. This AL query will not provide any label information, hence a waste of labeling cost. In fact, the occurrences of such disagreement queries can be common in that: (1) the model is inaccurate in the initial AL process due to low label rate, (2) and traditional AL selects most uncertainty node to label. Both make the model cannot accurately assess the likelihood of selected unlabeled instances belonging to the oracle’s KBS.
>
> In summary, the effectiveness of our design stems from the consideration of soft labels in relaxed queries and model training.
> We also appreciate your assessment about “The idea of soft-labeling is intriguing!”
>
> ### 2. Influence of the number of classes
>
> The reviewer raises a very good point. Based on the reviewer’s comment, we examine the model accuracies as a function of the number of classes on the ogbn-arxiv dataset. Specifically, we gradually increase the number of classes and the corresponding number of nodes in the ogbn-arxiv dataset, and report the mean test accuracy of 10 runs with and without the soft-label. As shown in the newly added Figure 7(a) in Appendix.J, although more classes increase the complexity of classification problem (thus a decreasing model accuracy), we see that the performance gain of adding soft label decreases slightly as the number of classes increases.
>
> The reason is that although the soft-max is taken over the remaining classes, the distribution of the output of soft-max (i.e., soft label) is expected to be highly skewed, which can help to alleviate the effect of increased class number on the quality of the soft label. To demonstrate this, we add a new Figure 7(b) showing the percentage of the node whose top-K accuracy is larger than 80\%. As the percentages of top K class increases (i.e., K/total class number), more than 90\% nodes only need the predicted top 20\% classes to guarantee the top-K accuracy larger than 80\%, which means few classes predominate the softmax distribution.

---

> > ### Comment · Reviewer_5Qmn · 2021-11-18
> > **Thanks**
> >
> > Thanks for your detailed response. Your feedback addresses my concerns.

---

### Decision · Program_Chairs · 2022-01-20

**Decision:**

Accept (Poster)

**Comment:**

This paper proposes a new approach to graph-based active learning, using the query whether the predictions made by the current model are correct or not.
Although the theoretical underpinnings of the proposed approach are a bit weak, the problem formulation that is newly proposed in this paper makes sense from a practical point of view, and the paper makes a simple and interesting proposal that would be worth sharing with the community.